# Changes in Vitellogenin, Abdominal Lipid Content, and Hypopharyngeal Gland Development in Honey Bees Fed Diets with Different Protein Sources

**DOI:** 10.3390/insects15040215

**Published:** 2024-03-22

**Authors:** Mustafa Güneşdoğdu, Aybike Sarıoğlu-Bozkurt, Ahmet Şekeroğlu, Samet Hasan Abacı

**Affiliations:** 1Department of Animal Production and Technologies, Faculty of Applied Sciences, Muş Alparslan University, 49250 Muş, Türkiye; 2Department of Biochemistry, School of Veterinary Medicine, Bursa Uludag University, 16059 Bursa, Türkiye; aybikesarioglu0@gmail.com; 3Department of Animal Production and Technologies, Faculty of Agricultural Sciences and Technologies, Niğde Ömer Halisdemir University, 51240 Niğde, Türkiye; ahmet.sekeroglu@ohu.edu.tr; 4Department of Animal Science, Faculty of Agriculture, Ondokuz Mayıs University, 55139 Samsun, Türkiye; samet.abaci@omu.edu.tr

**Keywords:** *Apis mellifera*, physiology, diet, feeding, physiology

## Abstract

**Simple Summary:**

The physiology and performance of individuals in a bee colony are directly related to the quality of nutrients available in the colony. A diet enriched with high-quality protein increases the immunity of the colony and the individual bees. In this study, we examined the effect of diets with different protein sources on dietary consumption, vitellogenin (Vg), abdominal lipid content, and hypopharyngeal gland development in adult worker bees. Our results indicate that the use of certain protein sources in bee diets can improve the parameters investigated.

**Abstract:**

Honey bees play an important role in the pollination of flowering plants. When honey bee colonies are deficient in pollen, one of their main nutrients, protein supplements are required. In this study, the effects of diets with six different protein sources on the physiological characteristics of worker bees (vitellogenin (Vg), abdominal lipid content (ALC), hypopharyngeal gland (HPG)) and consumption were investigated. The protein sources of the diets (diet I, …, diet VI) included pollen, spirulina dust (*Arthrospira platensis* Gomont), fresh egg yolk, lyophilized lactose-free skimmed milk powder, active fresh yeast, and ApiProtein. It was identified that consumption by worker bees was highest in the diet group supplemented with spirulina (diet II). Although there was no statistical difference regarding the Vg content in the hemolymph, numerically, the highest content was found in diet group IV (lyophilized lactose-free skimmed milk powder) (4.73 ± 0.03 ng/mL). ALC and HPG were highest in the group fed diet II. These results suggest that offering honey bees diets with certain protein sources can support their physiological traits.

## 1. Introduction

Honey bees (*Apis mellifera* L.) are the most important pollinator insects due to their crucial role in our ecosystem [1]. To prevent the complete death of a colony and ensure its healthy development, the needs of individual colonies should be known [2]. It is important for the health and performance of workers and queens that they are well nourished [3]. Nutrition is the intake of nutrients required by the body from the outside through the mouth, and honey bees also need nutrition to survive [4]. However, bee nutrition is highly dependent on the environment, particularly the composition of flora in the landscape, which can vary over time [5]. For honey bees to survive and reproduce, nutrients, including carbohydrates (nectar and honey), amino acids (pollen), lipids (fatty acids and sterols), vitamins, minerals (salt), and water [6], must be present in the colony [7]. Worker bees collect nectar (incl. honeydew honey), pollen, and water from nature and produce propolis from resources collected from nature. However, in different seasons, the amount of resources collected from nature varies depending on the needs of the colony [8]. Like other living creatures, bees need carbohydrates, whereby glucose is used as an energy source, and the surplus is stored as fat in the body [9].

Pollen, which is well known as the units formed in the male reproductive organs of flowering plants [10], is an important resource because it provides the bees with vital nutrients, including proteins, minerals, lipids, and vitamins [11]. The type and nutrient composition of pollen varies depending on the type and age of the plant, season, and climate. The protein content of pollen varies between 2.5% and 61% [12]. Grosh et al. [13] reported that a higher protein content of pollen is preferred by forager bees. The rearing of a larva requires 25 mg to 37.5 mg of protein. This corresponds to 125–187.5 mg of pollen. An adult bee consumes an average of 3.4–4.3 mg/day of pollen [14].

However, there are some proteins synthesized in the honey bee body that are important for structuring within the colony and hierarchical order [15]. One of these proteins is vitellogenin (Vg), a glycolipoprotein. Vitellogenin is the main source of storage proteins in the hemolymph. Also, Vg is a phosphoglycoprotein, with a molecular weight of 180 kDa, that is synthesized from the fat body. This protein can reach up to 80 μg/μL in winter bees, and it ensures the development of the fat body [16]. Vg synthesis depends on the availability and quality of pollen [17], and this lipoprotein has a direct effect on the onset of foraging behavior and life expectancy in bees [18]. Additionally, it is reported that there is more Vg in nurse bees than in forager bees [19]. Notably, the fat body is a dynamic tissue involved in lipid –carbohydrate metabolism, protein synthesis, amino acid and nitrogen metabolism, the endexine system, and the detoxification of nitrogen metabolism. The fat body plays an important role in energy storage and utilization in bees [20], and nurse bees have more abdominal fat than forager bees. The fat body increases with the protein nutrition from adult emergence. Also, the transition from nursing to foraging is associated with a reduction of about 50% in abdominal fat in a worker bee [21].

Generally, the nutrients derived from pollen are made available to the members of the colony by converting them into jelly-like royal jelly and brood food in the glands on the head of the worker bee. The conversion takes place in the hypopharyngeal glands in the bee’s head [22]. The size of the mandibular and hypopharyngeal glands (HPGs) of young nurse bees reaches its maximum between 6 and 10 days of age with pollen consumption [23]. Although the HPG is present in all the individuals of honey bee colonies, it is most developed in worker bees [24]. Recent studies have shown that the glands develop from the first day of the pupal stage [25]. Development is directly influenced by food availability [6]. When comparing the acini size of bees fed with pure bee pollen and protein supplements, supplements are more effective [26]. The HPG size differs in bees that are fed mono-floral pollen. Spring is the time when brood production begins, and the population increases in honey bee colonies [27]. In Turkey, honey bees show an increase in brood production from early spring to early July, when nectar and pollen flow are plentiful. After that, brood production gradually decreases, and from September they almost stop brood production, which is associated with a lack of food supply.

The aim of this study was to investigate the effect of different artificial feeds in Muş Province, the Republic of Türkiye, during the summer and fall seasons. Our focus was on the formulation of a pollen substitute diet, as natural pollen sources are decreasing [28]. The effects of these diets on worker bee consumption and the vitellogenin (Vg), abdominal lipid content, and head weight of bees were investigated.

## 2. Materials and Methods

### 2.1. Experimental Design—Apiaries, Colonies, and Hives

This study was conducted at Muş Alparslan University (38°46’15″ N 41°25’42″ E), Türkiye, utilizing the Langstroth hive, with a 10-frame capacity and a bottom board with a pollen trap. However, the pollen trap was not closed during the study, and so, natural pollen input was not hindered. The study involved 49 honey bee (*A.m. caucasica*) colonies in fixed apiary sites. There were six diet treatments and one non-diet treatment, with seven colonies per group. Also, during the study, queen bees were produced from the same breeding colony and replaced in June to minimize the differences due to genetics, nutrition, and age, as described by Pirk et al. [29]. Muş province has a continental climate due to its distance from the sea, and the temperature varies between −29 °C and +37 °C. Some climatic data of the coordinates given in the study are shown in Table 1 [30]. 

### 2.2. Preparation of Bee Diets

The raw materials and their ratios in the diets used to feed the bees in the experiment are given in Table 2. The only difference between diet I and diet II was the addition of spirulina to diet II. Equal amounts of pollen and substitute products were used in the other diets (Table 2). Canola oil was added according to the recommendation of Christopher Cutler et al. [31]. Canola oil was not added to diet III (fresh egg yolk). The reason was to ensure the non-fluid consistency of the diet and to avoid excessive fat content. ApiProtein is produced from inactive brewer’s yeast powder. It is a commercial product sold as a protein supplement in bee cake preparation. Poly-floral frozen pollen collected in spring was used in the diets. The diets were prepared and kept in a refrigerator at +4 °C until they were offered to the colonies. Colonies were given 500 g of each food, and during the experiment, the cakes that were not consumed in the colonies were replaced with new ones every week.

### 2.3. Chemical Analyzes of Diets

The percentages of crude protein, crude ash, crude fat, crude cellulose, and dry matter of the diets fed to the bee colonies in the experiment were determined, as described by Dumlu [32].

Dry matter (%): The moisture content of the diet samples was determined by drying to a constant weight using a redLINE moisture analyzer. Drying was carried out at 105 °C for 48 h with 5 g of the diet samples. The remaining weight was then calculated as the dry matter percentage.

Crude protein (%): The protein content was determined using the Kjeldahl method. Specifically, 0.20 g of ground diet samples were weighed and placed in digestion tubes. The samples were first digested with sulfuric acid in the presence of a catalyst. To accelerate the reaction in the combustion process, 2 g of catalyst tablets were added, and the combustion process was started. After the combustion process, the tubes were subjected to distillation with the Gerhardt apparatus, and then, the amount of hydrochloric acid consumed was determined by titration, and the amount of nitrogen was determined. The nitrogen percentage obtained was multiplied by 5.6 to determine the amount of protein. 

Crude fat (%): The diet samples were dried in an oven at 50 °C, and then, 2 g was placed on filter paper. The samples were then placed in the oil extraction apparatus for 6 h, after which the oils were evaporated, and the remaining samples were weighed and calculated.

Crude ash (%): 5 g of the diet samples were placed in sterile crucibles and incinerated in a Heraeus muffle furnace at 550 °C for 5 h. The residue was weighed, and the percentage of ash content was determined.

Crude cellulose (%): Samples were oven-dried (50 °C), sieved through a 1 mm diameter sieve, weighed to 0.5 g, and placed in F57 bags. After the bags were placed in an ANKOM device, they were boiled first with sulfuric acid and then with an alkaline sodium hydroxide solution (NaOH). The bags were then placed in a 250 mL beaker and soaked in acetone for 3 to 5 min. Then, the acetone was removed by squeezing the F57 bags and leaving them on a bench until the acetone evaporated. The F57 bags were then weighed, placed in crucibles, and then dried in an oven at 105 °C for 2–4 h. The crucibles taken from the oven were cooled in a desiccator and weighed (A1 = bag + fiber + crucible). The crucible + bags were incinerated in a muffle furnace set at 600 ± 15 °C for 2 h. In the end, the crucibles were taken to the desiccator, cooled, and weighed (A2 = crucible + ash). The residue content of the empty bag was also calculated (A3 = crucible + empty bag). Then, the crude cellulose percentage was calculated as follows:$$Crude\;Cellulose\;(\%)=\frac{100\times \left[\left(A1-A2\right)-\left(A3-A2\right)\right]}{\left[\left(A1-A2\right)\right]-\left(A3\right)}$$

In the determination of the pH values, 2 g of the diet samples was dissolved in 15 mL of distilled water and measured after a waiting period of 24 h [33,34]. The nutrient content of the diets is given in Table 3.

### 2.4. Diet Consumption

The diets were given to each bee colony on frames in a nylon refrigerator bag. To minimize the loss of water from the diet, a hole was drilled in the bottom of the bag for the bees to pass through. Diet consumption was determined once per week for a total of 16 weeks and calculated using the following equation: consumption per period = initial diet weight (500 g)—final diet weight [33]. The bees had access to natural resources (pollen, water, nectar, propolis) *ad libitum*. 

### 2.5. Vg Protein Analysis

For the analysis, 100 adult worker bees collected from each colony in the first week of November were transported to the laboratory under cold chain conditions, and stored at −18 °C. The vitellogenin (Vg) content of 10 nurse bees from different feeding groups was determined using the ELISA method. A procedure described by Mayack and Naug [35] was used for hemolymph extraction. Adult bees were killed by freezing, and the mouths of the tubes were sealed to prevent possible contamination of the hemolymph. For the analysis, the distal end of the bee antennae was cut off with scissors, and each bee was placed upside down in a centrifuge tube for 30 s at 16,000× *g*. Thereafter, 2 mL of the hemolymph dripping from the antennal tip by centrifugation was diluted with 58 mL of distilled water. The collected samples were analyzed for Vg content using an ELISA kit (VGT-MBS 284624, MyBioSource, San Diego, CA, USA) [36]. Since there is no specific Vg ELISA kit for honey bees yet, the kit prepared for the closest relative, *Nosonia vitripensis*, was used.

### 2.6. Abdominal Lipid Content

To determine the abdominal lipid content, a simple method, involving washing the abdomen with ether [37], was used. For this purpose, fifteen adult worker bees were collected from each experimental group. Their abdomen was separated from the body, the intestines were removed, and their weight (mg) was determined after drying at 45 °C for 72 h (FDAW). Then, the abdomens were placed in a tube and shaken in a shaker for 24 h with the addition of 4.5 mL of ethyl ether. The abdomens were then dried for a second time at 45 °C for 3 days, and their weights were determined for a second time (AWSD) [36]. The abdominal lipid content (mg/g) was calculated as follows:$$ALC\left(mg/g\right)=(\frac{FDAW}{AWSD})\times 1000$$where:ALC: abdominal lipid content;FDAW: first drying abdomen weight;AWSD: abdomen weight after second drying.

### 2.7. Hypopharyngeal Gland (HPG) Development

The bees obtained from each experimental group were collected in glass tubes and transported to the laboratory in a cool box. In the laboratory, the heads were separated from the thorax, and the wet and dry head weights were determined. For each measurement, 20 adult worker bees per colony were used, and the process was repeated 3 times. Thus, a total of 60 adult worker bees per colony were used on the same date. The dry weight was determined after the heads were held for 24 h in a 60 °C oven [33]. The dry weight was calculated using the formula described below [37]:$$W\left(g\right)=\frac{W_{hw}}{N}\times 1000$$
where:W: the weight of a single worker bee’s head (mg);W_hw_: the weight of all bees’ heads (g);N: the number of weighted bees.

### 2.8. Statistical Analysis

The normality assumption of the data was examined with the Kolmogorov–Smirnov test, and it was determined that the normality assumptions were met for the experiment. (*p* > 0.05). Statistical analyses were performed with one-way analysis of variance (ANOVA). A value of 0.05 was considered significant in the analysis of variance (ANOVA) to determine significant differences. Differences between groups were assessed with Duncan’s multiple-range test. The mean and standard error of the mean (SEM) values are given as descriptive statistics. The SPSS package program was used (SPSS Inc., Chicago, IL, USA). 

## 3. Results

### 3.1. Consumption

The results for diet consumption are indicated in Table 4. It was found that total weekly consumption and consumption over the entire study period were significantly different among the experimental groups (*p* < 0.01). The diet consumption of the colonies fluctuated continuously during the study period. The overall mean diet consumption was highest between August and September, while the lowest consumption was recorded in October. When comparing the diet consumption during the study period, the highest consumption was found for diet II with the addition of spirulina (*Arthrospira platensis*). Diet I with the addition of fresh flower pollen (diet I) ranked second with the highest consumption. The lowest consumption was recorded for diet III with the addition of fresh egg yolk. Diets IV, V, and VI were consumed in approximately equal amounts.

### 3.2. Hemolymph Vg Values

There was no statistical difference among the experimental groups regarding the Vg values (ng/mL) (*p* > 0.05; Figure 1). Although there was no statistical difference, the highest value of vitellogenin was found in group IV (freeze-dried lactose-free skim milk powder) (4.73 ± 0.03 ng/mL), and the lowest value was found in worker bees in colonies fed diet III (fresh egg yolk) (4.66 ± 0.01 ng/mL). More Vg was detected in the non-fed control group than in diet III, diet V, and diet VI groups.

### 3.3. Abdominal Lipid Content (ALC)

Table 5 shows that there was a significant difference among the experimental groups in terms of the ALC of adult worker bees (*p* < 0.05). The highest ALC (0.68 ± 0.04 mg/bee) was identified in bees fed diet II (spirulina). Bees fed diet VI had the second highest ALC (0.45 ± 0.05 mg/bee). These values are significantly higher than those of the other groups. The overall mean ALC was 0.26 ± 0.33 mg/bee. The control group had the lowest ALC. Also, the time–diet interaction effect on ALC was statistically insignificant (*p* > 0.05; Table 5).

### 3.4. HPG Development

There was a statistical difference among treatments regarding the wet head weight of adult worker bees (*p* < 0.05; Table 6). The highest mean wet head weight was found in the diet II (Spirulina) and diet VI (ApiProtein) groups. The lowest wet head weight was determined in the diet III (fresh egg yolk) group. The wet head weight was highest in all groups on 8/8/2022. The overall mean wet head weight was 12.49 ± 0.55 (mg/bee) (Table 6). It was observed that the time–diet interaction effect was insignificant regarding the wet weight of the heads (*p* > 0.0; Table 6). Also, the dry head weight of adult worker bees differed among the treatments (*p* < 0.05; Table 7). The highest dry head weight was recorded for those fed diets II, IV, and VI. Although the wet head weight of diet IV was significantly lower than that of the other diets, there was no difference regarding the dry head weights. The overall mean dry head weight was 5.66 ± 0.11 (mg/bee) (Table 7). Furthermore, the highest dry head weight among the sampled dates was recorded on 7/9/2022. In addition, the time–diet interaction effect was significant for the dry head weight (*p* < 0.05; Table 7).

## 4. Discussion

### 4.1. Diet Consumption

Beekeepers often feed their colonies with artificial and commercial pollen substitutes to compensate for the lack of pollen sources in nature. This supplementation is known to increase the bee population by stimulating brood production [38,39]. The palatability and nutritional value of the feed have a direct effect on the production and productivity of the offspring [40]. The experimental results presented in this study provide valuable information on the response of colonies (consumption) and individual honey bees (Vg, HPG, and ALC) to different pollen substitute feeds. Consumption is an important indicator for understanding the quality and effects of a diet and can be influenced by several characteristics (colony and diet) [33]. Our results showed that the pollen substitute diet prepared with fresh pollen and spirulina was consumed to a higher extent by the bees during the experimental period. The feed enriched with egg yolk was only eaten to a limited extent by the bees. Haydak [4] reported that bee cakes prepared with egg yolk, which contained about 30% protein, were readily consumed by the bees. The highest consumption was found in the diet III (date paste) group at 213.2 g/colony [41]. Gemeda [42], Israr et al. [43], Islam et al. [44], and Aly et al. [45] reported weekly consumptions of bee cake of 384.9, 49.53, 71.90, and 47.42 g/colony, respectively. Amro et al. [46] reported that the highest consumption was recorded for bee cake prepared with brewer’s yeast. Taha (2015) reported that the highest consumption was recorded in the cake prepared with a honey–pollen mixture. Ricigliano et al. [47] reported that increasing the amount of spirulina in the diet led to a decrease in consumption. De Grandi-Hoffman et al. [26] reported that in bee colonies fed pollen and pollen substitute (MegaBee), the consumption of pollen cake was higher than that of the substitute.

### 4.2. Hemolymph Vg Values

Adequate nutrient availability is required for Vg production, and the level of circulating Vg in the hemolymph is closely related to protein supply and quality [17]. Honey bees fed with pollen-containing forage show higher Vg gene expression than bees that do not receive forage [42]. Nelson et al. [19] and Corby-Harris et al. [48] reported that Vg levels are high in guard bees and low in forager bees and that pollen consumption increases these levels. Amdam et al. [49] reported an increase in endogenous juvenile hormone levels and a decrease in vitellogenin levels (0–1 μg/μL) during the transition from keeper to forager bees. They also reported that the decrease in Vg levels led to an increase in the rate of taste in worker bees. Almeida-Dias et al. [50] reported vitellogenin levels of 54.29, 12.14, 32.24, and 39.46 μg/μL. In a similar study, this value was reported as 2.5–3.0 ng/mL. In another study, it was reported that the Vg content of colonies fed *Papaver somniferum* (poppy), *Cistus creticus* (pink spruce), poly-floral pollen, bee cake, and sugar syrup in the fall season and control group colonies ranged from 3.8 to 4.0, 4.0 to 4.2, 4.2 to 4.4, 3.8 to 4.0, 4.0 to 4.4, and 3.8 to 4.0 ng/mL [36].

### 4.3. Abdominal Lipid Content

The lipid content in the abdomen of honey bees is influenced by many factors. For example: pyruvate carboxylase, cytosolic malate dehydrogenase, and ATP–citrate lyase. In addition, lipid synthesis is directly related to the age of the bees [21]. The fact that nurse bees lose body volume before they become forager bees indicates that biosynthetic activity is reduced in the fat body. Corby-Harris et al. [51] found no correlation between the amount of food ingested and the fat body. However, they found that a low fat body occurred with a poor quality food source. The fat body is responsible for immunity, the production of antimicrobial molecules, the regulation of the endocrine system, and the storage of nutrients in bees [20]. Since fat storage in the body is related to nutrition, it is assumed that a high-quality diet leads to a fatter body [52]. In a similar study, it was reported that the amount of fat body was equal in adult bees fed dry spirulina and multifloral pollen [52]. When comparing bees fed with algae powder from *Azolla pinnata* and corn pollen, the bees fed with algae powder had larger fat bodies [36]. Yeast-enriched diets contain proteins and lipids that are necessary to increase the amount of abdominal fat body [53]. In a study conducted in Estonia, Italy, Spain, and Sweden, Vanderplanck et al. [54] reported the amount of fat body as 0.11, 0.13, 0.9, and 0.9 mg, respectively.

### 4.4. HPG Development

The higher dry bee head weight reflects the development of the glands in the head of the bees [22]. The hypopharyngeal gland is one of several glands in young worker bees that is responsible for the secretion of royal jelly, along with the mandibular gland, the cerebral gland, and the mammary gland [55]. Several studies have shown a positive correlation between dry weight and lifespan [17]. El Ghbawy et al. [36] reported a dry head weight of 8.28, 9.72, and 8.07 (mg/bee) in the Azolla patty, pollen patty, and control groups, respectively. Es’kov and Es’kov [56] reported that the wet head weight of adult bees ranged from 9.1 to 11.9 mg. In a study like ours, the wet and dry weights of adult bees were reported to be 9.5–11.5 mg/bee and 3.6–4.0 mg/bee, respectively [57]. The worker bees were found to have lower wet and dry weights than the pollen-fed bees when they were not fed pollen until the 7th day after cell emergence [58].

## 5. Conclusions

This study was able to determine that the addition of spirulina, fresh flower pollen, and lyophilized lactose-free skim milk powder to the diets of bees can enhance the consumption and some physiological characteristics of honey bees. However, the addition of egg yolk to diets depresses the above traits in honey bees. Generally, the results of this study give clues on some benefits of the formulated protein substitutes; however, further studies are recommended on the topic.

## Figures and Tables

**Figure 1 insects-15-00215-f001:**
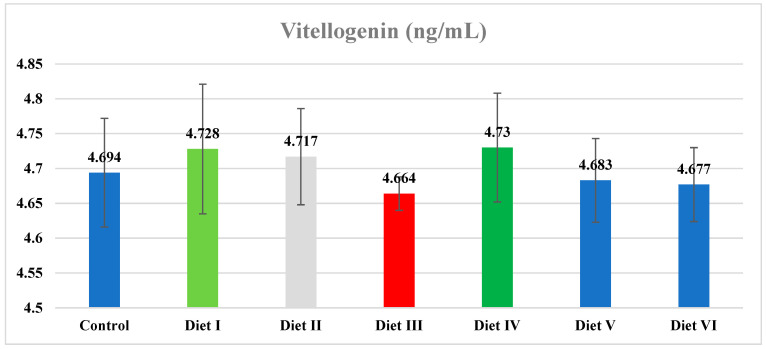
Vg content/values of groups in late fall.

**Table 1 insects-15-00215-t001:** Climate data of Muş Province.

Climatic Characteristics (2022)	July	August	September	October	November
Monthly cloudless days	29	27	22	12	10
Monthly total sunshine duration	334.4	289.6	230.7	188.7	137.8
Average pressure (hPa)	863.5	864.7	867.7	871.7	872.2
Average relative humidity (%)	27.1	24.8	33.5	51.5	72.5
Average wind speed (m/s)	0.9	1.0	0.9	0.9	0.7
Min. temperature (°C)	15.2	14.5	9.2	3.8	−0.1
Max. temperature (°C)	38.1	38.3	33.9	26.4	18.6
Average temperature (°C)	26.1	27.2	18.9	13.5	7.2
Avg. precipitation (mm = kg/m^2^)	1.0	0.0	16.8	25.4	39.8

hPa: hectopascal, m/s: meter/second, °C: centigrade degree, mm = kg/m^2^: millimeter = kilogram/square meter.

**Table 2 insects-15-00215-t002:** Raw materials and their ratio in the various diet treatments.

Components	Diets					
	Diet I	Diet II	Diet III	Diet IV	Diet V	Diet VI
P (g)	250	250	125	125	125	125
PS (g)	250	250	250	250	250	250
Sp (g)	-	2	-	-	-	-
FEY (g)	-	-	125	-	-	-
LSLMP (g)	-	-	-	125	-	-
FY (g)	-	-	-	-	125	-
Ap^®^ (g)	-	-	-	-	-	125
EISS (mL)	150	150	50	150	150	150
AVMC (g)	1	1	1	1	1	1
CO (mL)	10	10	-	10	10	10
Piece ($)	1.12	1.19	1.40	1.71	1.54	1.26

P: pollen, PS: powdered sugar, Sp: spirulina dust (*Arthrospira platensis* Gomont), FEY: fresh egg yolk, LSLMP: lyophilized skim lactose-free milk powder, FY: active fresh yeast, Ap^®^: ApiProtein^®^, EISS: enzymatic invert sugar syrup, AVMC: amino acid–vitamin–mineral complex for bees, CO: canola oil.

**Table 3 insects-15-00215-t003:** Chemical composition of different honey bee diets (%).

Diets	Crude Ash (%)	Crude Fat (%)	Crude Protein (%)	Crude Cellulose (%)	pH	Dry Matter (%)
Diet I	0.81	9.33	6.97	1.24	3.40	84.84
Diet II	0.88	10.12	8.41	1.49	3.54	85.86
Diet III	0.59	9.62	6.36	0.74	2.31	85.56
Diet IV	1.17	9.63	6.65	1.05	4.67	86.04
Diet V	0.73	11.52	6.85	1.74	3.88	84.31
Diet VI	1.18	7.93	11.04	2.23	4.20	84.97

**Table 4 insects-15-00215-t004:** Consumption of different pollen substitutes (g/colony/week; mean ± SEM).

Diets	N	16.07–13.08.2022	13.08–10.09.2022	10.09–8.10.2022	8.10–29.10.2022	Overall Mean
Diet I	7	287.00 ± 5.98 c	478.25 ± 5.59 a	466.53 ± 7.31 a	442.69 ± 10.91 a	418.59 ± 15.27 B
Diet II	7	464.14 ± 5.76 a	484.96 ± 2.16 a	483.64 ± 6.42 a	460.35 ± 5.66 a	473.27 ± 3.27 A
Diet III	7	90.00 ± 5.92 d	138.71 ± 9.21 d	23.57 ± 2.66 d	All Dead	84.09 ± 10.86 D
Diet IV	7	436.07 ± 6.56 a	425.89 ± 3.16 b	235.03 ± 26.06 c	60.82 ± 11.78 c	289.45 ± 30.51 C
Diet V	7	303.14 ± 9.69 bc	347.42 ± 22.17 c	274.75 ± 6.71 b	152.17 ± 8.47 b	269.37 ± 15.27 C
Diet VI	7	328.10 ± 11.96 b	370.57 ± 12.37 c	273.07 ± 7.73 b	153.32 ± 7.19 b	281.26 ± 16.40 C
Overall mean	49	318.07 ± 19.22 AB	374.30 ± 18.76 A	292.76 ± 25.56 B	253.85 ± 27.94 C	299.17 ± 15.26
*p*-Value		<0.001	<0.001	<0.001	<0.001	<0.001

The difference between the means given with the different letters in the same column is statistically significant (*p* < 0.05), SEM: standard error of mean, N: sample size.

**Table 5 insects-15-00215-t005:** Abdominal lipid content of adult bees at different measurement times (mg/bee; mean ± SEM).

Diets	N	9.07.2022	30.07.2022	20.08.2022	10.09.2022	Overall Mean
Diet I	7	0.12 ± 0.01 b	0.20 ± 0.04 c	0.12 ± 0.03 bcd	0.11 ± 0.01 b	0.14 ± 0.01 C
Diet II	7	0.75 ± 0.07 a	0.74 ± 0.11 a	0.60 ± 0.08 a	0.62 ± 0.08 a	0.68 ± 0.04 A
Diet III	7	0.15 ± 0.02 b	0.18 ± 0.03 c	0.22 ± 0.05 bcd	All Dead	0.16 ± 0.03 C
Diet IV	7	0.15 ± 0.04 b	0.11 ± 0.02 c	0.11 ± 0.01 cd	0.11 ± 0.03 b	0.12 ± 0.01 C
Diet V	7	0.12 ± 0.04 b	0.12 ± 0.02 c	0.25 ± 0.04 bc	0.11 ± 0.01 b	0.15 ± 0.02 C
Diet VI	7	0.55 ± 0.15 a	0.47 ± 0.09 b	0.28 ± 0.07 b	0.48 ± 0.09 a	0.45 ± 0.05 B
Control	7	0.17 ± 0.06 b	0.18 ± 0.04 c	0.07 ± 0.02 d	0.04 ± 0.02 b	0.11 ± 0.02 C
Overall mean	49	0.29 ± 0.04	0.28 ± 0.03	0.24 ± 0.03	0.23 ± 0.04	0.26 ± 0.33
*p*-Value		<0.00	<0.001	<0.001	<0.001	<0.001
T × D Int.		0.373				

The difference between the means given with the different letters in the same column is statistically significant (*p* < 0.05), SEM: standard error of mean, N: sample size: T: time, D: diet, ×: interaction.

**Table 6 insects-15-00215-t006:** Wet head weights of adult worker bees (mg/bee; mean ± SEM).

Diets	7.07.2022	21.07.2022	8.08.2022	23.08.2022	7.09.2022	21.09.2022	Overall Mean
Diet I	12.04 ± 1.05 bc	10.51 ± 1.24 bc	12.25 ± 1.24 bc	11.37 ± 0.84 c	10.97 ± 0.72 c	11.25 ± 0.39 b	11.40 ± 0.75 BC
Diet II	18.31 ± 0.85 a	19.55 ± 1.23 a	19.50 ± 1.08 a	18.58 ± 0.69 a	17.72 ± 0.87 a	17.68 ± 1.01 a	18.56 ± 0.85 A
Diet III	8.08 ± 0.47 d	8.57 ± 0.38 c	9.58 ± 1.43 c	8.26 ± 0.53 d	All Dead	All Dead	8.62 ± 0.48 D
Diet IV	10.61 ± 0.078 c	10.74 ± 1.09 bc	11.38 ± 1.04 bc	11.32 ± 1.16 c	11.54 ± 0.90 bc	11.57 ± 1.04 b	11.19 ± 0.92 C
Diet V	11.97 ± 0.63 bc	12.84 ± 1.23 bc	13.71 ± 1.34 bc	13.11 ± 0.77 bc	12.88 ± 1.08 bc	12.50 ± 0.89 b	12.83 ± 0.94 BC
Diet VI	13.90 ± 0.91 b	14.38 ± 1.86 b	14.60 ± 1.78 b	14.74 ± 1.30 b	14.11 ± 1.18 b	13.92 ± 1.21 b	14.27 ± 1.31 B
Control	11.17 ± 1.07 c	12.57 ± 1.83 bc	12.50 ± 1.25 bc	10.51 ± 1.05 cd	10.71 ± 0.79 c	10.72 ± 0.83 b	11.36 ± 1.10 BC
Overall mean	12.30 ± 0.52	12.74 ± 0.67	13.36 ± 0.63	12.56 ± 0.55	12.20 ± 0.53	11.78 ± 0.64	12.49 ± 0.55
*p*-Value	<0.001	<0.001	<0.001	<0.001	<0.001	<0.001	<0.001
T × D	0.111						

The difference between the means given with the different letters in the same column is statistically significant (*p* < 0.05), SEM: standard error of mean, T: time, D: diet, ×: interaction.

**Table 7 insects-15-00215-t007:** Dry head weights of adult worker bees (mg/bee; mean ± SEM).

Diets	7.07.2022	21.07.2022	8.08.2022	23.08.2022	7.09.2022	21.09.2022	Overall Mean
Diet I	5.70 ± 0.49	5.37 ± 0.42 ab	5.67 ± 0.28 abc	5.63 ± 0.35 abc	5.71 ± 0.23 b	5.76 ± 0.43 ab	5.64 ± 0.28 AB
Diet II	5.36 ± 0.50	6.17 ± 0.38 a	6.25 ± 0.29 a	6.25 ± 0.26 ab	6.11 ± 0.33 ab	6.28 ± 0.23 ab	6.07 ± 0.22 A
Diet III	5.50 ± 0.41	5.30 ± 0.32 ab	5.18 ± 0.35 bc	5.42 ± 0.44 bc	All Dead	All Dead	5.35 ± 0.27 AB
Diet IV	5.90 ± 0.30	6.24 ± 0.33 a	6.08 ± 0.20 ab	6.21 ± 0.27 ab	6.11 ± 0.21 ab	6.24 ± 0.24 ab	6.13 ± 0.21 A
Diet V	5.81 ± 0.37	5.80 ± 0.15 ab	5.58 ± 0.29 abc	5.16 ± 0.33 c	5.74 ± 0.32 b	5.60 ± 0.31 ab	5.61 ± 0.26 AB
Diet VI	6.11 ± 0.24	5.98 ± 0.32 ab	5.75 ± 0.25 abc	6.58 ± 0.31 a	6.75 ± 0.30 a	6.42 ± 0.18 a	6.27 ± 0.12 A
Control	4.84 ± 0.48	4.93 ± 0.36 b	5.03 ± 0.31 c	5.17 ± 019 c	4.80 ± 0.24 c	4.80 ± 0.21 b	4.92 ± 0.25 B
Overall mean	5.60 ± 0.15	5.68 ± 0.13	5.65 ± 0.11	5.77 ± 0.13	5.79 ± 0.13	5.44 ± 0.23	5.66 ± 0.11
*p*-Value	0.414	0.066	<0.047	<0.010	<0.001	<0.001	<0.001
T × D	<0.006						

The difference between the means given with the different letters in the same column is statistically significant (*p* < 0.05), SEM: standard error of mean, T: time, D: diet, ×: interaction.

## Data Availability

All data are stated in this manuscript.

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
