# Peer review of "Changes in Vitellogenin, Abdominal Lipid Content, and Hypopharyngeal Gland Development in Honey Bees Fed Diets with Different Protein Sources"

_insects, 2024, doi:10.3390/insects15040215_

Round 1

Reviewer 1 Report

Comments and Suggestions for Authors

I find the manuscript very interesting for the field. However, there are some issues with experimental design and some details which missing in methodology.

Conclusions section needs to be revised, it is not written to underline results of this study and some general statements need to be removed from it.

The tittle also needs to be revised. What does mean “Honey Bee Groups”?

I am not a native speaker, but it seems to me that English should be improved and also some terminology seems not completely adequate for scientific papers from honey bee-science field.

Line 39: individual colonies? Do you mean individual bees or colonies?

Line 47: what is “direct proportion” please explain.

Line 48: Bees doesn’t collect propolis, they produce it from resources collected from nature.

Line 67-68: “More Vg is present in bees that live in colonies than in 67 bees that do not live in colonies” please explain this with more information.

Line 97: it is not written correct “7 colonies per treatment x 6 feed treatments x 1 non-feed treatment” I think it is 7 colonies per treatment x (6 feed treatments + 1 non-feed treatment)

Lines 98-100: “As Pirk et al., [29] stated, all queen bees were produced from the same breeding colony and renewed in June in order to minimize the differences due to genetics, nutrition, and age.” You cannot cite others statements in your methodology/experimental design. Please add description of your methods and design.

Line 103: was the pollen trap in bottom board used/closed and for how long during the year.

Line 104: the altitude range 1300-2950 m of the whole region is less important than the altitude of the certain location of apiaries.

Lines 113-114: “Canola oil is not added to the diet” it is not clear if you added or not canola oil, please revise this.

Line 154: “The abdominal lipid content is calculated using different methods” Did you use different methods or just one which you described. Please revise this.

Line 190: “p=0.000” it is impossible to be exactly 0. Please check this with statistician.

Conclusion section needs to be rewritten. You have to conclude your results not to write general statements about beekeeping nowadays (e.g. We are increasingly feeling the effects of climate change; Especially in the fall season, high-quality feeding has a positive effect on the overwintering ability of bee colonies) moreover you did not analyzed overwintering in this study so please remove this statements about overwintering from Conclusion section as well as from simple summary and abstract.  

Moreover try to write conclusions to be self-explanatory and that reader could clearly understand it. E.g. diet II, diet I, and diet VI does not mean anything, add at least most important characteristics of diets in conclusion.

Comments on the Quality of English Language

I am not a native speaker, but it seems to me that English should be improved and also some terminology seems not completely adequate for scientific papers from honey bee-science field.

Author Response

Please refer to the uploaded file regarding our responses to the comments. 

Reviewer 2 Report

Comments and Suggestions for Authors

The title of the article is well reflected. The introduction is short and to the point. In the materials and methods section, it is good to describe the quantity of additives included in the food. Eating additional protein has a stimulating effect on the development of the hypopharyngeal glands. The glands are very well observed under a microscope during the spring development of the bee colonies. The presented study shows with photographic material the stimulating effect of a herbal extract on the development of the glands ( Shumkova, R., R. Balkanska. 2021. Influence of IMMUNOSTART HERB feeding on the bee colonies devel­opment. Bulgarian Journal of Agricultural Science, 27 (5), 896–902.)

I would like to know if the authors have done more extensive research under the microscope?

I advise if they have measured the amount of pollen in the hive during the experiment to add this essential information. Replace it with meteorological data.

I support this kind of research because it is important in the nutrition of bee colonies.

Author Response

(The authors gave the same response as above.)

Round 2

Reviewer 1 Report

Comments and Suggestions for Authors

They corrected all the issues. Manuscript is now ready for publication.